# Synthesis of Reduced Graphene Oxide/Titanium Dioxide Nanotubes (rGO/TNT) Composites as an Electrical Double Layer Capacitor

**DOI:** 10.3390/nano8110934

**Published:** 2018-11-09

**Authors:** John Paolo L. Lazarte, Regine Clarisse Dipasupil, Gweneth Ysabelle S. Pasco, Ramon Christian P. Eusebio, Aileen H. Orbecido, Ruey-an Doong, Liza Bautista-Patacsil

**Affiliations:** 1Chemical Engineering Department, De La Salle University, 2401 Taft Avenue, Manila 1004, Philippines; aileen.orbecido@dlsu.edu.ph; 2Chemical Engineering Department, Malayan Colleges Laguna, Pulo-Diezmo Rd. Cabuyao City, Laguna 4025, Philippines; clarissedipasupil@gmail.com (R.C.D.), gwenethspasco@gmail.com (G.Y.S.P.); 3Chemical Engineering Department, University of the Philippines Los Baños, College, Laguna 4031, Philippines; rpeusebio@up.edu.ph; 4Institute of Environmental Engineering, National Chiao Tung University, Hsinchu 30010, Taiwan; radoong@mx.nthu.edu.tw; 5Department of Biomedical Engineering and Environmental Sciences, National Tsing Hua University, Sec. 2 Kuang-Fu Road, Hsinchu 30013, Taiwan

**Keywords:** reduced graphene oxide, titanium dioxide nanotubes, electrical double layer capacitor, electrode, nanocomposite

## Abstract

Composites of synthesized reduced graphene oxide (rGO) and titanium dioxide nanotubes (TNTs) were examined and combined at different mass proportions (3:1, 1:1, and 1:3) to develop an electrochemical double layer capacitor (EDLC) nanocomposite. Three different combination methods of synthesis—(1) TNT introduction during GO reduction, (2) rGO introduction during TNT formation, and (3) TNT introduction in rGO sheets using a microwave reactor—were used to produce nanocomposites. Among the three methods, method 3 yielded an EDLC nanomaterial with a highly rectangular cyclic voltammogram and steep electrochemical impedance spectroscopy plot. The specific capacitance for method 3 nanocomposites ranged from 47.26–165.22 F/g while that for methods 1 and 2 nanocomposites only ranged from 14.03–73.62 F/g and 41.93–84.36 F/g, respectively. Furthermore, in all combinations used, the 3:1 graphene/titanium dioxide-based samples consistently yielded the highest specific capacitance. The highest among these nanocomposites is 3:1 rGO/TNT. Characterization of this highly capacitive 3:1 rGO/TNT EDLC composite revealed the dominant presence of partially amorphous rGO as seen in its XRD and SEM with branching crystalline anatase TNTs as seen in its XRD and TEM. Such property showed great potential that is desirable for applications to capacitive deionization and energy storage.

## 1. Introduction

Electrical double layer capacitors (ELDCs) are important engineering devices that can be fabricated using specific nanomaterials. EDLCs are mainly used in energy storage where they function as electrodes in two poles—positive and negative. With their ability to store charge/ions, EDLCs have likewise been applied in water treatment technologies such as capacitive deionization [1].

Unlike other energy storage devices such as pseudocapacitors, EDLCs are made from nanomaterials that do not react with and cause redox reaction among the ionic species present in the solution. When no reaction happens, ions between them maintain their identity as desired and arrange themselves on the electrode surface in response to the electric field between the two electrodes to form an electrochemical double layer. This layer consists of ionic species that accumulated on the electrode–electrolyte interface, effectively making two layers of capacitors connected in series in each of the electrode surface [2]. As explained by the Stern-Guoy-Chapman model, the model currently used for EDLCs, electrolytes close to an EDLC electrode form two distinct layers called the Helmholtz layer and diffuse layer. The Helmholtz layer contains ions adsorbed on the surface of the electrode bounded by two planes: the inner Helmholtz plane (IHP) and outer Helmholtz plane (OHP). On the other hand, the diffuse layer contains nonspecifically adsorbed ions. This behavior cannot just be achieved by any nanomaterial as it requires high material inertness, conductivity, and surface area available for capacitance [3]. In order to assess whether a material can be an EDLC electrode, these properties may be determined through electrochemical means such as cyclic voltammetry (CV) and electrochemical impedance spectroscopy (EIS). For an EDLC material, a rectangular CV plot with high specific capacitance should be obtained, as this indicates minimized redox reactions during charge transfer, and a steep linear EIS Nyquist plot should be obtained, as this indicates low material resistance [4,5,6].

Among the nanomaterials available, carbon is known to meet EDLC characteristics. With its many allotropes and morphological variations, this chemically and thermally inert substance can be synthesized to have a large surface area that is highly conductive. Several carbon-based compounds such as activated carbon, carbon aerogel, reduced graphene oxide, and their modifications are considered EDLC nanomaterials [6,7,8,9,10]. One modification that could be done to achieve and enhance desired properties would be to synthesize carbon sheets spaced with oxides [3]. This is expected to transpire in a composite of reduced graphene oxide and titanium dioxide as these compounds have electron-rich carbons and polar functional groups that enhance conductivity and stabilize the composite formed. Furthermore, geometric conversion of TiO_2_ particles into different forms such as nanobelts or nanorods improves EDLC characteristics due to larger rGO spacing caused by nanobelts as reported in the work of Xiang et al. [11]. The performance of this composite is enhanced further when TiO_2_ is converted to nanotubes instead of nanobelts or nanorods as reported in the work of Gobal and Faraji [12]. The improved performance of this composite is due to the larger rGO spacing caused by the hollow annular region of titanium dioxide nanotubes (TNTs) compared to the spacing caused by more compact nanobelts (NBs). Although these studies achieved material improvement with TiO_2_ structure modification, insights over the effect of wider range of proportions and different reaction approaches in the composite properties were not covered.

Thus, in this work rGO and TNT composites were (1) synthesized using three different methods at different proportions and (2) assessed based on cyclic voltammetry (CV) and electrochemical impedance spectroscopy (EIS). Lastly, to acquire a better understanding of the outcome of the electrochemical tests in relation to molecular, chemical, and microscopic make-up of the composite, the best composites were (3) characterized using XRD, SEM, and TEM and compared to pure rGO and TNT characteristics.

## 2. Materials and Methods

### 2.1. Materials and Reagents

The following chemicals and reagents were used: graphite (graphite powder, Aldrich, Milwaukee, WI, USA), sulfuric acid (95–98% H_2_SO_4_, Merck, Darmstadt, Germany), phosphoric acid (85% H_3_PO_4_, Sigma, St. Louis, MO, USA), potassium permanganate (KMnO_4_ powder, Merck, Darmstadt, Germany), hydrogen peroxide (30% H_2_O_2_, Riedel-de Haën, Sleeze, Germany), hydrochloric acid (36.5–38% HCl, J.T. Baker, Phillipsburg, NJ, USA), sodium borohydride (98% NaBH_4_, Aldrich, Milwaukee, WI, USA), titanium dioxide (TiO_2_ ST01 powder, Ishihara Sangyo, Tokyo, Japan), sodium hydroxide (99% NaOH pellets, J.T. Baker, Phillipsburg, NJ, USA), ethanol (99.5–98% C_2_H_6_O, Riedel-de Haën, Sleeze, Germany), deionized (DI) water, carbon black (carbon powder, Uni-onward Corp., New Taipei, Taiwan), N-methyl-2-pyrrolidone (>99% NMP, Sigma-Aldrich, Milwaukee, WI, USA), polyvinylidenefluoride (PVDF, Aldrich, Milwaukee, WI, USA), and sodium sulfate (99% Na_2_SO_4_ powder, Merck, Darmstadt, Germany). All reagents were used as received without further purification.

### 2.2. Synthesis of Reduced Graphene Oxide (rGO)

Synthesis of rGO proceeded in two steps: the conversion of graphite to graphene oxide (GO), and the conversion of GO to rGO [13,14]. For synthesis of GO, the reagents used were graphite (2 g), H_2_SO_4_ (225 mL), H_3_PO_4_ (25 mL), and KMnO_4_ (6 g). The mixture was stirred for 12 h at 45 °C, poured into 225 mL DI water and kept in ice bath. While stirring, 3 mL of H_2_O_2_ was added. The mixture was then allowed to cool down. The cooled mixture separated into different tubes were equally treated with 1 M HCl and pH 11 phosphate buffer. Then the samples were washed with DI water and centrifuged for 20 min at 9000 rpm until the pH reached 5–7. Afterwards, 10 mL ethanol was added each sample and was dried to obtain a film of GO. This GO (300 mg) was again dispersed in water (300 mL) by ultrasonication for 3 h for conversion to rGO. The dispersed GO was reduced with NaBH_4_ (1800 mg) using an oil bath at 100 °C for 12 h. The same method of washing was employed.

### 2.3. Synthesis of Titanium Dioxide Nanotubes (TNT)

Synthesis of TNT was based on microwave method [15]. TNTs were prepared by mixing 300 mg TiO_2_ ST01 into 10 mL of 10 M NaOH using ultrasonication. The mixture was reacted using a microwave (CEM, Matthews, NC, USA) for 3 h at 150 °C under 600 W initial power, then washed with 1 M HCl and ethanol until neutral, and dried in oven at 50 °C for 12 h.

### 2.4. Synthesis of Reduced Graphene Oxide/Titanium Dioxide Nanotubes (rGO/TNT) Composites

There were three methods tested to produce a composite of rGO and TNT tagged as Method 1, 2 and 3.

Method 1: Sodium borohydride reduction of graphene oxide mixed with titanium dioxide nanotubes

The synthesized graphene oxide (GO) and titanium dioxide nanotubes (TNT) were prepared at five different proportions: 100%, 75%, 50%, 25% and 0% GO. The mixture was then reacted with NaBH_4_ to reduce GO as discussed in rGO synthesis. These samples were labelled as GO/TNT composites.

Method 2: Microwave synthesis of titanium dioxide nanotubes with reduced graphene oxide

The synthesized reduced graphene oxide (rGO) was combined with titanium dioxide (TiO_2_) at five different proportions: 100%, 75%, 50%, 25% and 0% rGO. The mixture was reacted in microwave to convert TiO_2_ to TNT for 3 h as discussed in TNT synthesis. These were samples labelled as rGO/TiO_2_ composites.

Method 3: Microwave combination of titanium dioxide nanotubes and reduced graphene oxide

The synthesized reduced graphene oxide (rGO) and titanium dioxide nanotubes (TNT) were combined using a microwave for 3 h at 150 °C and 600 W initial power at five different proportions: 100%, 75%, 50%, 25% and 0% rGO. These were labelled as rGO/TNT composites.

### 2.5. Characterization and Electrode Performance Assessment

Prior to electrochemical analysis with cyclic voltammetry (CV) and electrochemical impedance spectroscopy (EIS), the samples were prepared at a composition of 70% active material (nanocomposite), 20% carbon black, and 10% PVDF. The mixture was powdered and dissolved in NMP for coating. The amount of NMP in milliliters used was around 5 times the mass of the sample in milligrams. The mixture was subjected to 2-hour ultrasonication then coated on an inert current collector. The mass of the current collector before coating and after drying of the coat were recorded. The synthesized electrodes were electrochemically characterized using Autolab PGSTAT302N (Metrohm Autolab BV, Utrecht, The Netherlands), a potentiostat/galvanostat instrument capable of controlling the input parameters of CV and EIS. For CV, the following parameters were used: scan rates of 5, 10, 20, 50, 100, 200 mV/s and a potential window from −0.8 to 0.2 V. In CV, a rectangular plot with a high specific capacitance indicates that the nanomaterial follows EDLC behavior. The specific capacitance was computed using the formula:(1)$$SC=\frac{\int IdV}{2\nu mV}$$
where *I* = response current density (A/cm^2^), *V* = potential window (V), *v* = potential scan rate (mV/s), and *m* = mass of the active material in the electrode (g). For EIS, the frequency range to generate the impedance was from 100,000 to 0.01 Hz. In EIS, a steep and straight line is observed if the nanomaterial follows EDLC behavior.

### 2.6. Electrode Characterization and Performance

All nanocomposites synthesized from the best method were tested with x-ray diffraction (XRD, Bruker, Bremen, Germany), and the best nanocomposite was further tested with a scanning electron microscope with energy-dispersive x-ray spectroscopy (SEM-EDS, Jeol, Ltd., Tokyo, Japan and Zeiss, Oberkochen, Germany) and transmission electron microscope (TEM, Jeol, Ltd., Tokyo, Japan). The pure materials GO, rGO, and TNT were analyzed using XRD and SEM to serve as reference and verification that correct starting materials were synthesized.

## 3. Results and Discussion

### 3.1. XRD, SEM, and CV of Precursors

As shown in Figure 1, the XRD of GO had a sharp peak at 10° as expected [13]. This peak corresponds to (001) diffraction plane and 0.848 nm interlayer spacing caused by oxygenated functional groups in the carbon allotrope such as hydroxyls, epoxides, carbonyls, and carboxyls [14,16]. Furthermore, the XRD of synthesized pure rGO recorded peaks at 24° and 43°. This corresponds to the literature with a phase shift of around 3° to the left as the expected peaks corresponding to rGO are found at 27° and 46°. Phase shifts for XRDs normally happen especially when different diffractometers are used. Phase shifts hold valid as long as all peaks shift to the same extent and direction.

The 24° peak corresponding to (002) diffraction plane indicates that reduction was successful while the 43° peak corresponding to (100) diffraction plane indicates high extent of reduction with interlayer spacing of 0.38 nm and 0.36 nm respectively [11,17,18].

The XRD of the synthesized titanate nanotubes recorded peaks at around 25°, 28°, and 48° as shown in Figure 2. This titanate phase is identified to be H_2_Ti_3_O_7_ [19]. The phase formed was not completely converted into TNT anatase phase desired for the composite synthesis in this study. Furthermore, the bulging behavior of the plot implies that the compound formed is amorphous in contrast to the desired crystalline behavior of anatase TNT [15]. However, as will be discussed in Section 3.5, the material improved during composite development of some samples.

SEM images obtained show that rGO and TNT were successfully synthesized. As shown in Figure 3A, sheets of rGO are found to be stacked together as expected [14]. Furthermore, the obtained image of TNT in Figure 3B showed randomly arranged tubes typical to hydrothermal reaction path in contrast to highly organized arrayed TNTs synthesized using electrochemical means [12,19].

As it was verified from XRD and SEM that the right materials were synthesized, their CV characteristics were then tested. As shown in Figure 4B, rGO behaves more ideally than its GO precursor in Figure 4A. Its CV plot at 200 mV/s is rectangular even at high scan rates unlike GO. This degree of rectangularity is already similar to EDLC materials in literature [6,7,8,9,10]. Moreover, TNT CV plot is rectangular, but it curves near the vertex potentials as shown in Figure 4C. These irregular shapes suggest that the electron transfer involves electrochemical reaction.

### 3.2. Electrochemical Performance of GO/TNT Composites

As shown in Figure 5, CV of composites from this method revealed that all three GO/TNT composites showed a variable flow of current as the potential changes from −0.8 to 0.2 V and vice-versa especially at 200 mV/s. Current variation was more distinct near the vertex potentials. This behavior was more pronounced in 1:1 and 1:3 GO/TNT composites.

The poor shape of the CV curves of the samples are in line with the low specific capacitance values obtained reported in Table 1. The composites all had lower specific capacitance than the pure materials themselves. Among all the samples, 3:1 GO/TNT had the highest specific capacitance among the three while the 1:1 GO/TNT sample had the lowest.

In terms of consistency, the specific capacitance variation shown in Figure 6 illustrates that 1:1 GO/TNT had more stable performance at higher scan rates. Nevertheless, its specific capacitance was the lowest. On the other hand, reduced graphene oxide showed a very sharp drop in specific capacitance at a low scan rate but had a very stable performance at higher scan rates. Compared with the specific capacitance of other materials at 5 mV/s, the specific capacitance of pure rGO is much better than the rest even if the lowest specific capacitance of rGO at 200 mV/s is around 85 F/g only. The samples in between had a behavior intermediate of the two extremes.

Moreover, the EIS profile of the samples as shown in Figure 7 still showed that the composites had a much higher resistance as evident from their significantly wider *x*-axis coverage compared to the pure materials, leading to a less steep line. The resistance at higher frequencies (first few points of the plot) was already at 50–100 Ω, which increased to 150–700 Ω as the frequency decreased. EDLCs are expected to have rapidly increasing imaginary impedance (*y*-axis value) with slowly increasing resistance (*x*-axis value) as points are recorded, starting from the highest frequency to the lowest frequency. Hence, a steep linear plot is desired. Thus, these findings together with CV results point out that composites synthesized using method 1 do not qualify as an EDLC nanomaterial. Failure to obtain an EDLC behavior from the composite implies that reduction of GO mixed with TNT is highly inhibited. Alternatively, the NaBH_4_ used in reducing GO might be detrimental to the TNT structure supported by oxygen functional groups as it is a strong reducing agent.

### 3.3. Electrochemical Performance of rGO/TiO_2_ Composites

As shown in Figure 8, CV of the composites formed from this method show that rGO/TiO_2_ composites behaved more closely to EDLC behavior compared to GO/TNT composites. At 200 mV/s, sample 1:1 rGO/TiO_2_ exhibited the most rectangular behavior among the three. Although the curve of 3:1 rGO/TiO_2_ is not as rectangular, it still possessed a good shape unlike 1:3 rGO/TiO_2_. Sample 1:3 rGO/TiO_2_ had bulges along the middle of the curve, implying a current fluctuation during the potential variation. This is a sign that the material still behaves like a pseudocapacitive material rather than an EDLC material.

As shown in Table 2, although these samples had a better CV in terms of shape, their specific capacitance was not able to surpass pure rGO, although samples 3:1 rGO/TiO_2_ and 1:3 rGO/TiO_2_ were able to surpass TNT in this respect.

The profile of specific capacitance with changing scan rate in Figure 9, show that rGO is still better than the rest even at its highest scan rate although it had a very high drop initially. The rest of the samples were more stable and had roughly the same rate of change except for 3:1 rGO/TiO_2_ which had a highly decreasing specific capacitance with increasing scan rate.

Further assessing the performance of the composites using EIS as shown in Figure 10 revealed that while rGO/TiO_2_ had a resistance that only ranged from 10–25 Ω at higher frequencies, significantly less than the resistance of GO/TNT, rGO and TNT still outperformed the composites with their steeper line. This signifies that as frequency is reduced, the composites had a greater increase in resistance compared to the two pure materials. Although this method showed improvement over an uninhibited reduction of graphene oxide as it was reduced before composite combination, this method still does not yield the desired EDLC. In this method, TNT formation failure is the probable cause of unsuccessful synthesis of an EDLC composite. This failure prevented the rGO sheets to be spaced by TNT.

### 3.4. Electrochemical Performance of rGO/TNT Composites

As shown in Figure 11, the CV of rGO/TNT composites showed improved behavior over the pure substances and composites synthesized previously. Among the CV of the composites, CV of 3:1 rGO/TNT had a highly rectangular shape even at 200 mV/s. Moreover, CV of 1:3 rGO/TNT possesses a slightly rectangular curve compared to that of 1:1 rGO/TNT. Sample 1:1 rGO/TNT had the least rectangular curve, implying that this proportion was unsuccessful in preventing electrochemical reaction from happening in the system.

Among all the samples, it is evident that sample 3:1 rGO/TNT composite had the highest specific capacitance with a magnitude of 165.22 F/g, as shown in Table 3. It was able to surpass even the rGO synthesized in this work. This was followed by 1:3 rGO TNT. The other composite, 1:1 rGO/TNT, failed to outperform even pure TNT which is reported to be incapable of behaving as an EDLC as TNT is a pseudocapacitive material.

Moreover, since specific capacitance changes with scan rate, an important measure to affirm that 3:1 rGO/TNT is the best composite would be its ability to maintain a high value of specific capacitance even at high scan rates. The trend of specific capacitance with scan rate change in Figure 12 show that 3:1 rGO/TNT, rGO, and 1:3 rGO/TNT would always be better than TNT and 1:1 rGO/TNT at any scan rate. Moreover, it is evident that 3:1 rGO/TNT could not be outperformed by any other material even by rGO at any given scan rate, making it the best composite in terms of its ability to attract ions.

Further analysis using EIS show that the performance of the composites was excellent with its relatively low resistance even at low frequencies as shown in Figure 13. Although not as small as the resistance of rGO/TiO_2_ composites, the high frequency resistance ranged only from 10–40 Ω. At lower frequencies, low resistance rate of increase was evidenced by the closeness of slopes of the composites to pure materials. Furthermore, among all composites, 3:1 rGO/TNT had the best performance as it gave high impedances unlike rGO, TNT, and 1:3 rGO/TNT. The 3:1 rGO/TNT composite was able to achieve this while keeping resistance low unlike 1:1 rGO/TNT, which had a significant increase of resistance that reached 40 Ω at the high frequency region that further increased at lower frequencies while 3:1 rGO/TNT only had a resistance of 20 Ω at that region. The findings in both CV and EIS for this composite reveals that this method was able to produce an EDLC material even better than the standard rGO at 3 rGO: 1 TNT mass proportion.

### 3.5. SEM, TEM, and XRD of rGO/TNT

As the rGO/TNT method produced the best composites, especially 3:1 rGO/TNT, they were characterized for a deeper understanding of their behavior. The XRD of all rGO/TNT composites is shown in Figure 14. This figure shows that the 23° broad peak of the amorphous rGO was concealed by the 25° sharp peak of TNT due to the relatively higher crystallinity of anatase [20]. Aside from this sharp peak corresponding to (101) diffraction plane and the 48° peak corresponding to (200) diffraction plane as found in the previously synthesized TNT, small peaks emerged at 38°, 54°, 55°, and 63° corresponding to (004), (105), (211), and (204) diffraction planes respectively [15,19,21]. The emergence of these peaks indicates complete conversion of amorphous titanate to crystalline anatase phase, suggesting that the material formed with method 3 yielded the desired form of each component, successfully avoiding the inhibitions caused by synthesizing composites using method 1 and 2. As pointed out earlier, the presence of TNT during the reduction of GO in method 1 may have competed with GO in consuming NaBH_4_, thereby inhibiting rGO formation. Although TNT is said to have an increased conductivity when reduced by NaBH_4_ due to oxygen vacancies, reduction of GO is still preferred. On the other hand, the presence of rGO in the formation of TNT in method 2 may have attracted the hydroxyl intermediates of TNT to its polar groups, preventing growth and assembly of the intermediates to form nanotubes [22].

With the knowledge that rGO/TNT composites had the same identity, the superior performance of 3:1 rGO/TNT could be qualitatively understood with the microscopic morphology provided by SEM-EDS and TEM. In the SEM images of the composite, the presence of rGO sheets is dominant as shown in Figure 15A,B while TNT is not visible. This implies that TNT was concealed within the rGO sheets. On the other hand, the presence of TNT is confirmed by the TEM images of the sample in Figure 15C,D. TEM images of the sample showed that the sample has a sheet-like structure and tube-like morphs embedded in it.

Furthermore, EDS analysis of the samples show that 3:1 rGO/TNT composite contains 62.34% carbon, 26.87% oxygen, 0.21% sodium, and 10.57% Ti. This validates the fact that TNTs were just covered at the rGO/TNT SEM image. Additionally, mass composition of TNT by EDS was 60.39% O, 38.94% Ti, and 0.67% Na. Taking the proportion of Ti and O from this data and obtaining the equivalent oxygen content attached to TNT, the computed composition of the nanocomposite is 73% rGO and 27% TNT or 2.7:1 rGO/TNT. This value is approximately equal to the initial reaction mixture of 3:1, confirming that the identity of each pure material (rGO and TNT) was preserved when they were combined.

The imaged structure and elemental analysis confirm that 3:1 rGO/TNT possesses the necessary characteristics for an EDLC as described by Bommier and Ji [3]. The morphology of 3:1 rGO/TNT allowed spacing in between rGO sheets that corresponds to a larger surface area. The high proportion of carbon allowed higher conductivity from carbon pi electron bonds. Moreover, the abundance of oxygen, a highly electronegative element, verifies the fact that the material is highly conductive. Lastly, the presence of polar molecules both in rGO and TNT enabled the material to be inertly bonded by strong intermolecular dipole-dipole forces. As a consequence of these properties, enhanced specific capacitance and minimized redox reactions were achieved by 3:1 rGO/TNT composite.

## 4. Conclusions

Based on the comparison of different methods of syntheses for rGO/TNT composites, it was evident that the best method would be to synthesize rGO and TNT separately before combining them. Synthesizing both materials separately was shown not to inhibit the formation of the other compound. With regard to proportions, the greatest presence of rGO in general was found to be superior even to pure rGO.

Among all the samples, 3:1 rGO/TNT possesses the desired EDLC characteristics as it was able to have the best performance in terms of CV curve shape, specific capacitance, and EIS Nyquist plot. Compared to studies found in literature, 3:1 rGO/TNT reported in this work achieved a specific capacitance value of at least 85 F/g even at 200 mV/s, while 7:3 rGO/TiO_2_ nanobelts only achieved a capacitance value of 56.2 F/g at 100 mV/s [11]. Furthermore, unlike the rGO/TNT composites available in literature, this composite did not have pseudocapacitive peaks in its CV plot that indicate redox reaction [12].

While TNT was expected to have the lowest specific capacitance compared to the rest since it is not a carbon-based material, it was responsible for obtaining the highest specific capacitance at 3:1 rGO/TNT composite. The presence of TNT allowed a greater surface area for ions to move within the rGO sheets as verified by SEM-EDS and TEM. Nevertheless, the presence of TNTs cannot be further increased as they affect the performance of rGO negatively as the influence of the highly conductive rGO is shielded with a greater presence of TNT.

With the desired EDLC behavior attained, 3:1 rGO/TNT will perform well in EDLC applications. It will have a better stability as an electrode as it prevents degradation of material due to redox reactions. The recommended future research directions to further examine this material would be to subject it to more molecular characterization tests such as BET and FTIR and to conduct experiments to test its applicability to common applications mentioned earlier such as energy storage and capacitive deionization.

## Figures and Tables

**Figure 1 nanomaterials-08-00934-f001:**
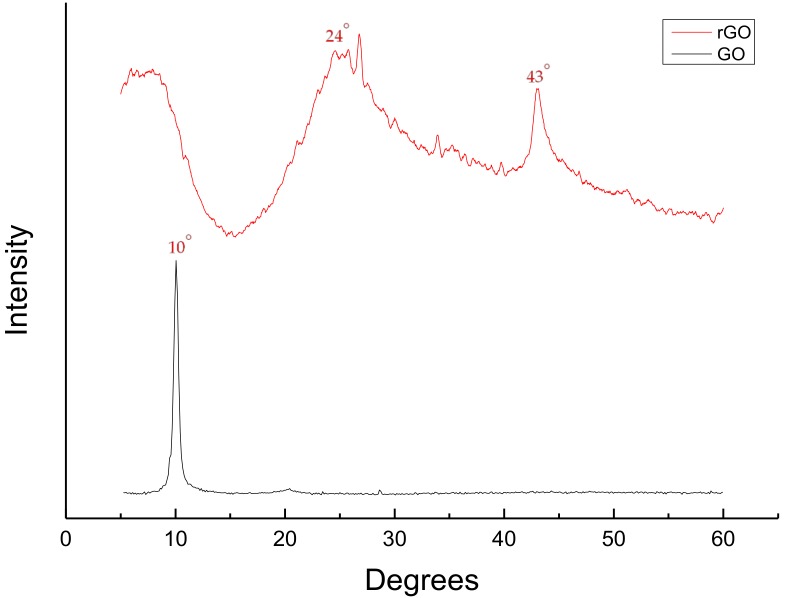
X-ray diffraction pattern of graphene oxide (GO) and reduced graphene oxide (rGO).

**Figure 2 nanomaterials-08-00934-f002:**
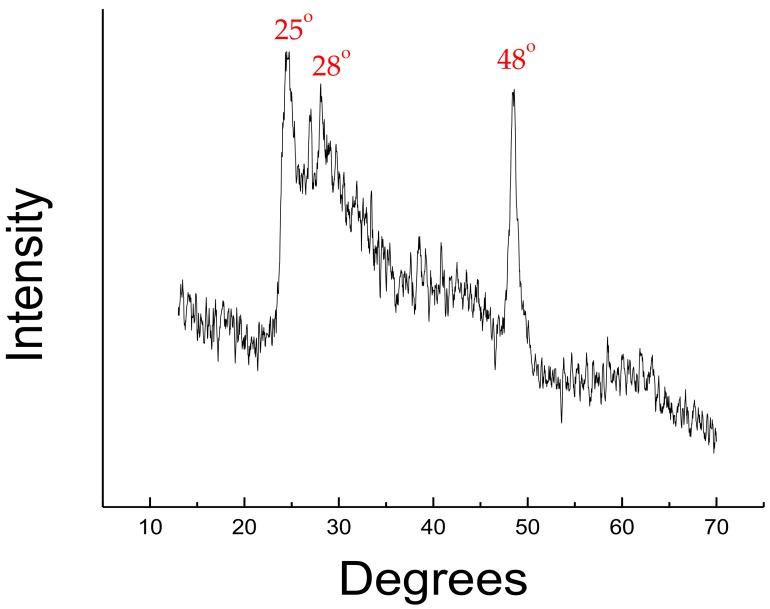
X-ray diffraction pattern of titanium dioxide nanotubes (TNT).

**Figure 3 nanomaterials-08-00934-f003:**
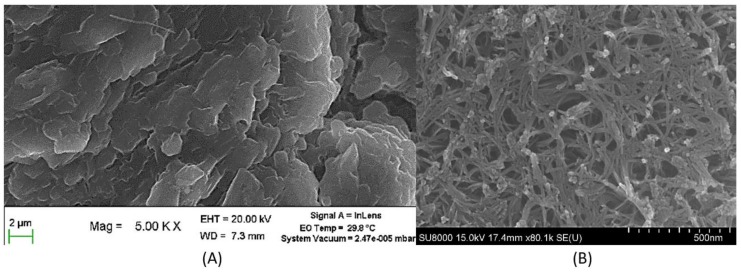
SEM image of pure (**A**) rGO and (**B**) TNT.

**Figure 4 nanomaterials-08-00934-f004:**
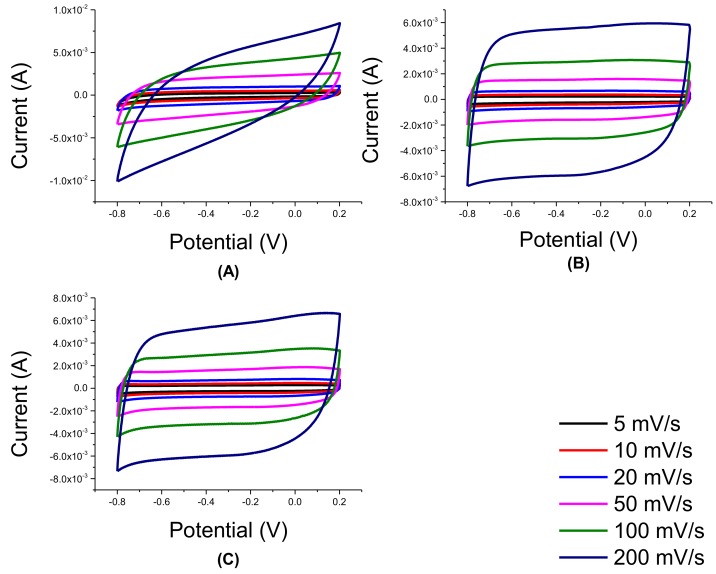
Cyclic voltammetry (CV) of pure materials: (**A**) GO, (**B**) rGO, and (**C**) TNT.

**Figure 5 nanomaterials-08-00934-f005:**
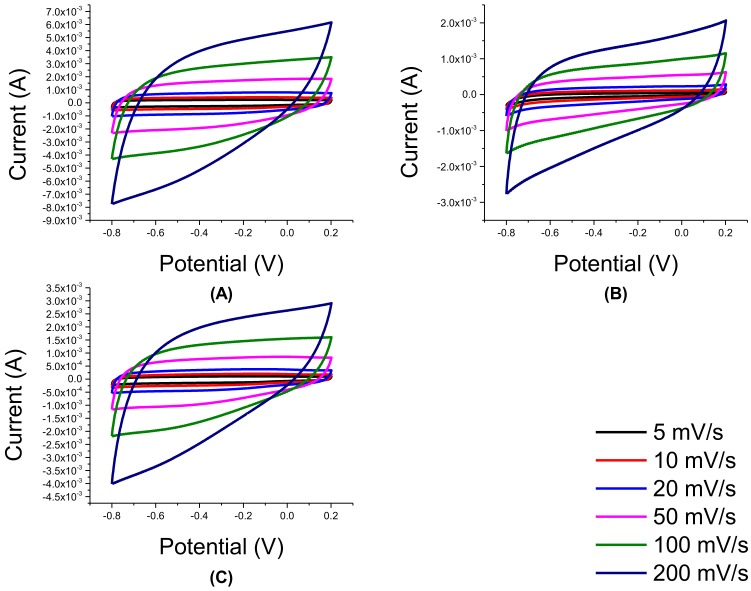
CV of GO/TNT: (**A**) 3:1, (**B**) 1:1, and (**C**) 1:3.

**Figure 6 nanomaterials-08-00934-f006:**
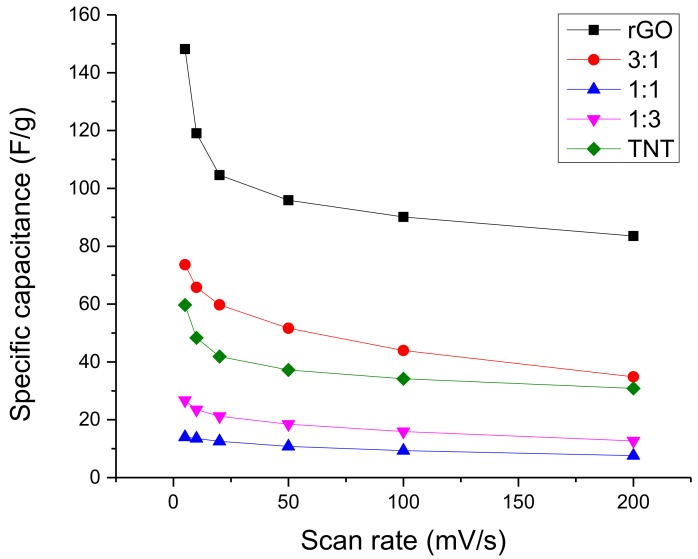
Specific capacitance variation with scan rate of GO/TNT composites.

**Figure 7 nanomaterials-08-00934-f007:**
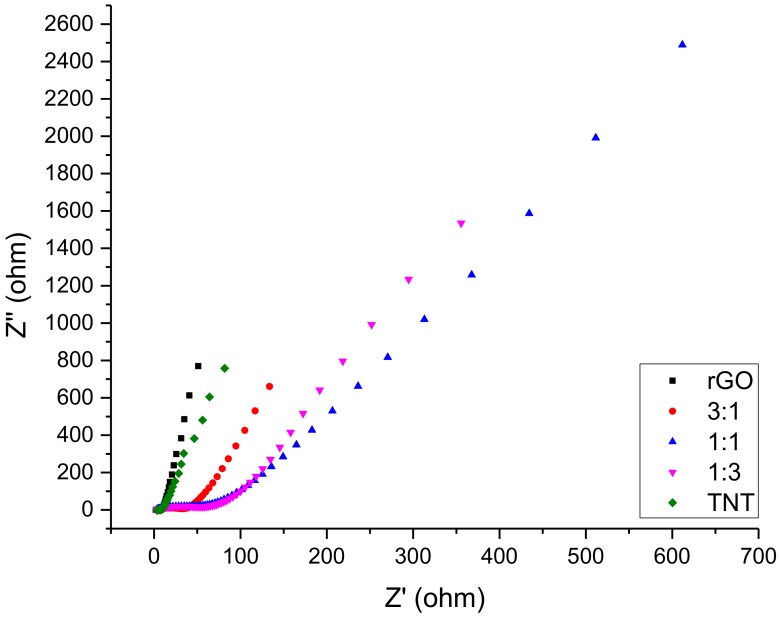
Electrochemical impedance spectroscopy (EIS) of GO/TNT composites.

**Figure 8 nanomaterials-08-00934-f008:**
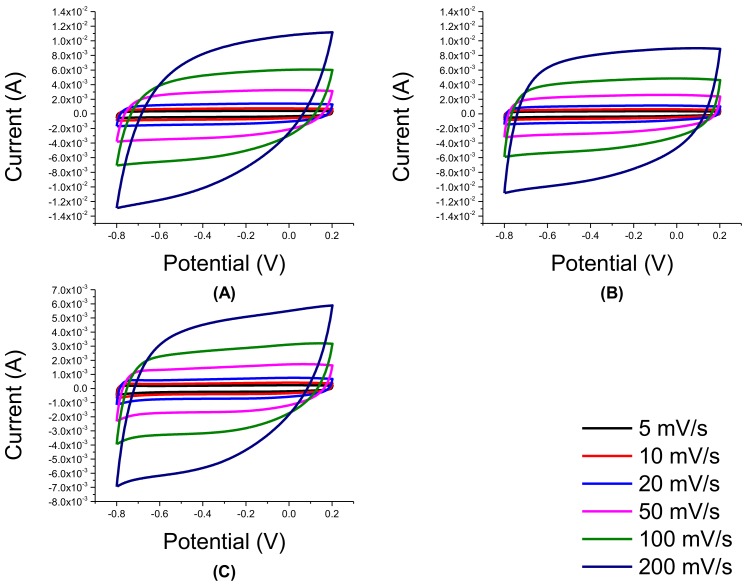
CV of rGO/TiO_2_: (**A**) 3:1, (**B**) 1:1, and (**C**) 1:3.

**Figure 9 nanomaterials-08-00934-f009:**
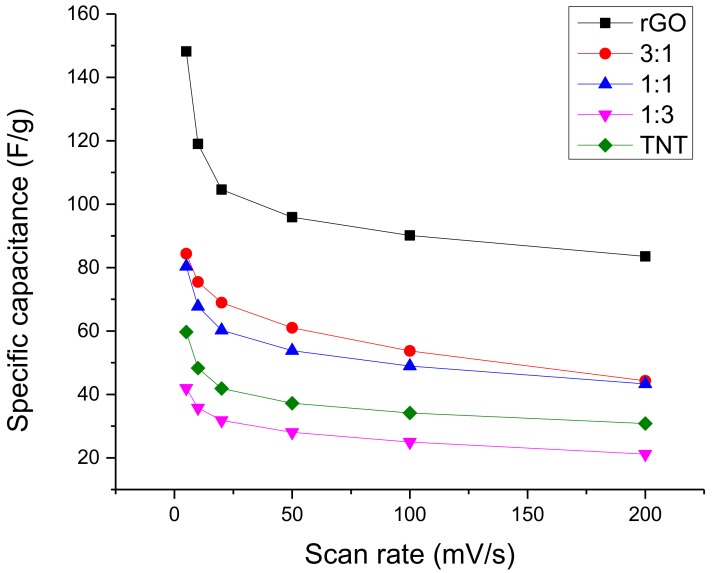
Specific capacitance variation with scan rate of rGO/TiO_2_ composites.

**Figure 10 nanomaterials-08-00934-f010:**
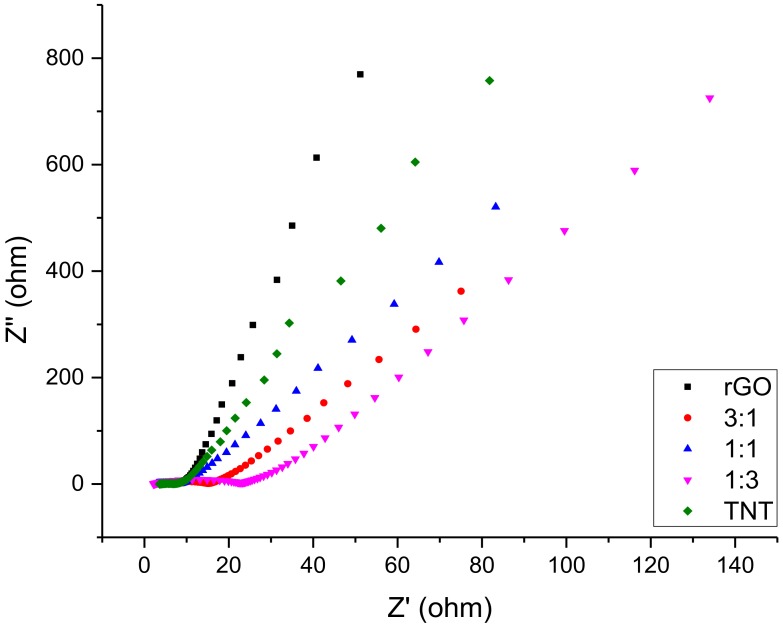
EIS of rGO/TiO_2_ composites.

**Figure 11 nanomaterials-08-00934-f011:**
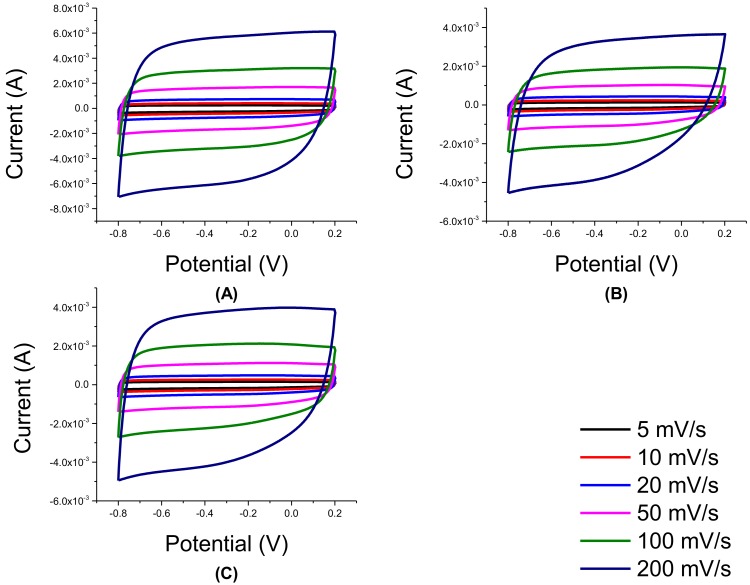
CV of rGO/TNT: (**A**) 3:1, (**B**) 1:1, and (**C**) 1:3.

**Figure 12 nanomaterials-08-00934-f012:**
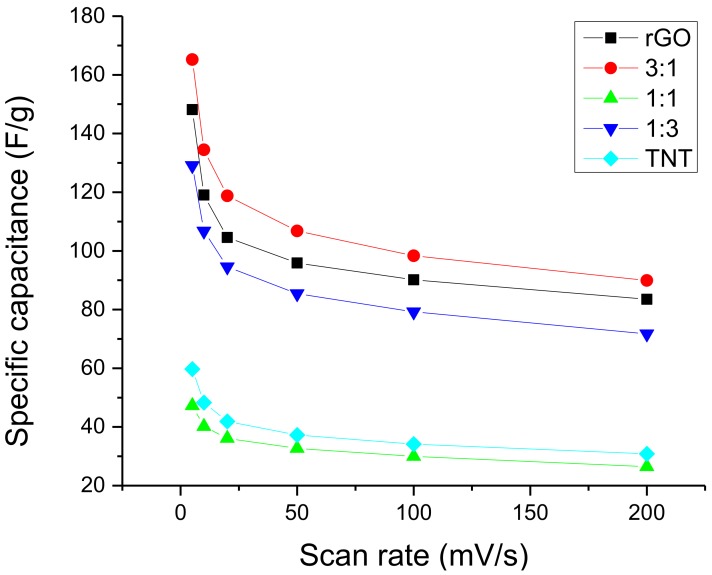
Specific capacitance variation with scan rate of rGO/TNT composites.

**Figure 13 nanomaterials-08-00934-f013:**
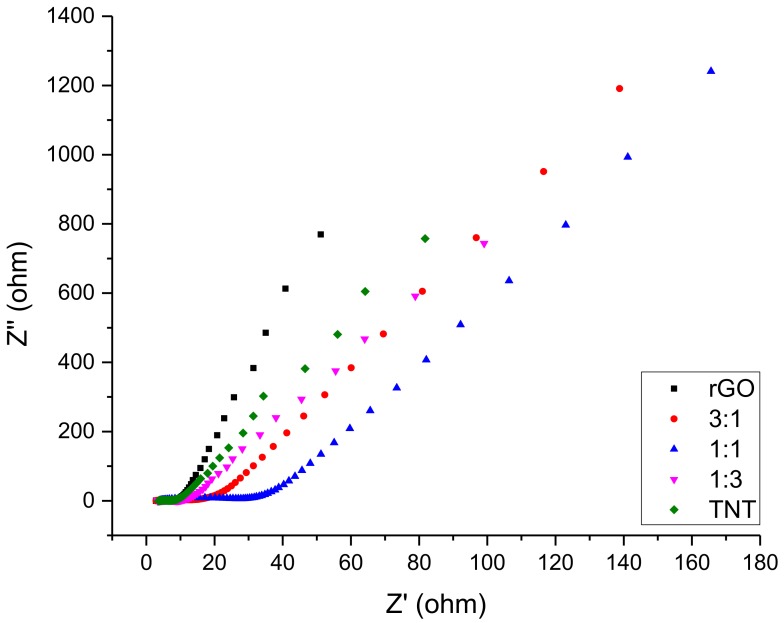
EIS of rGO/TNT composites.

**Figure 14 nanomaterials-08-00934-f014:**
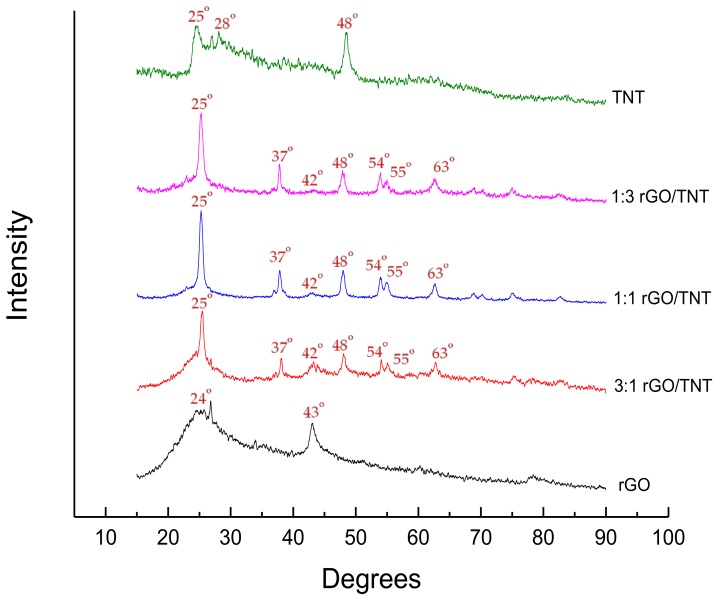
XRD of rGO/TNT composites and pure materials.

**Figure 15 nanomaterials-08-00934-f015:**
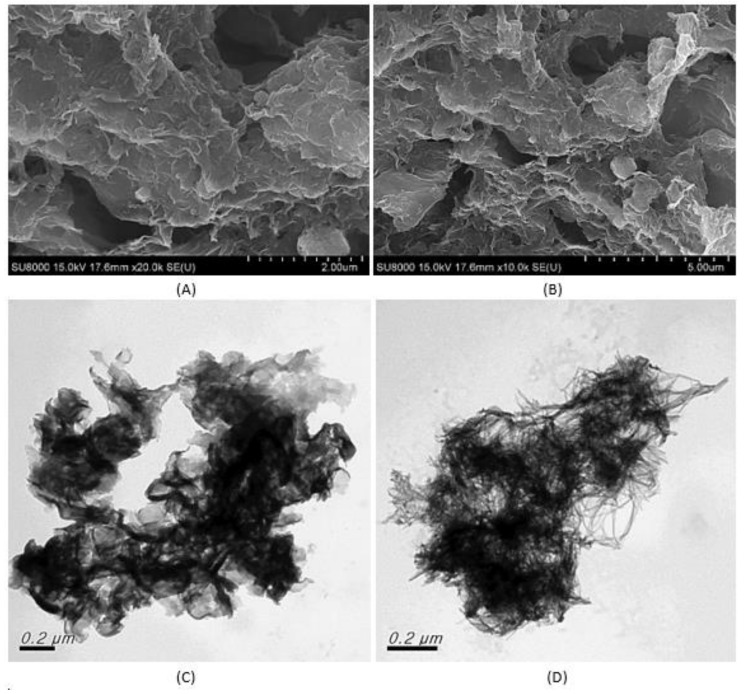
(**A**,**B**) SEM images and (**C**,**D**) TEM images of 3:1 rGO/TNT.

**Table 1 nanomaterials-08-00934-t001:** Specific capacitance of GO/TNT composites at 5 mV/s.

Material	Specific Capacitance (F/g)
rGO	148.18
3:1	73.62
1:1	14.03
1:3	26.69
TNT	59.69

**Table 2 nanomaterials-08-00934-t002:** Specific capacitance of rGO/TiO_2_ composites at 5 mV/s.

Material	Specific Capacitance (F/g)
rGO	148.18
3:1	84.36
1:1	80.36
1:3	41.93
TNT	59.69

**Table 3 nanomaterials-08-00934-t003:** Specific capacitance of rGO/TNT composites at 5 mV/s.

Material	Specific Capacitance (F/g)
rGO	148.18
3:1	165.22
1:1	47.26
1:3	129.08
TNT	59.69

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
