# Peer review of "Synthesis of Reduced Graphene Oxide/Titanium Dioxide Nanotubes (rGO/TNT) Composites as an Electrical Double Layer Capacitor"

_nanomaterials, 2018, doi:10.3390/nano8110934_

Reviewer 1 Report

This work presents an interesting study of the use of the reduced graphene oxide and titanium dioxide nanotubes in EDLCs with a special interest in the study of composite production approaches and ratios. However, the previous state of the art is not well documented in the present manuscript. This is not a new topic, and the authors need to clarify what is the incremental advance with respect to works such us:

Reduced graphene oxide/titanium dioxide composites for supercapacitor electrodes: shape and coupling effects, Journal of Materials Chemistry, 36, 2012.

Electrochemical synthesis of reduced graphene oxide/TiO2 nanotubes/Ti for high-performance supercapacitors,  June 2014, Ionics.

And references therein.

Along the text, EDLCs are referred to as nanomaterials. I guess that the authors should consider EDLCs as devices that can be partially fabricated based on certain nanomaterials.

In page 2, please provide vendors and characteristics of the graphite.

In page 3, please provide details on what Autolab PGSTAT302N is?

In page 8, please use symbols or arrows to differentiate clearly the curves in Figure 7. In Figure 8, the scale starting in -200 ohms is meaningless. Please use resistance symbol instead of impedance.

In page 13, the size of the font of the axis label is too strident.

Author Response

Response to Reviewer 1 Comments

 Point 1: This work presents an interesting study of the use of the reduced graphene oxide and titanium dioxide nanotubes in EDLCs with a special interest in the study of composite production approaches and ratios. However, the previous state of the art is not well documented in the present manuscript. This is not a new topic, and the authors need to clarify what is the incremental advance with respect to works such us:

 Reduced graphene oxide/titanium dioxide composites for supercapacitor electrodes: shape and coupling effects, Journal of Materials Chemistry, 36, 2012.

 Electrochemical synthesis of reduced graphene oxide/TiO2 nanotubes/Ti for high-performance supercapacitors, June 2014, Ionics.

 And references therein.

Response 1: A short discussion on the incremental advance of this work with respect to the related works enumerated above was included in the third paragraph of the Introduction (p. 2). The incremental advance was summarized as: “although these studies achieved material improvement with TiO2 structure modification, insights over the effect of wider range of proportions and different reaction approaches in the composite properties were not covered”.

 Furthermore, in the second paragraph of the Conclusion (p. 15), the following sentences were added: “Compared to studies found in literature, 3:1 rGO/TNT reported in this work achieved a specific capacitance value of at least 85 F/g even at 200 mV/s while 7:3 rGO/TiO2 nanobelts only achieved a capacitance value of 56.2 F/g at 100 mV/s [11]. Furthermore, unlike the rGO/TNT composites available in literature, this composite did not have pseudocapacitive peaks in its CV plot that indicate redox reaction[12].”

Point 2: Along the text, EDLCs are referred to as nanomaterials. I guess that the authors should consider EDLCs as devices that can be partially fabricated based on certain nanomaterials.

Response 2: The statement, “Electrical double layer capacitors (ELDCs) are important engineering nanomaterials”, in the first paragraph of the Introduction (p. 1) was changed to “Electrical double layer capacitors (ELDCs) are important engineering devices that can be fabricated using specific nanomaterials”.

Point 3: In page 2, please provide vendors and characteristics of the graphite.

Response 3: The following information were added in the Materials and Methods under Materials and Reagents (p. 2): “(graphite powder, Aldrich)”

Point 4: In page 3, please provide details on what Autolab PGSTAT302N is?

Response 4: The following description was added in Materials and Methods under Characterization and Electrode Performance Assessment: “…a potentiostat/galvanostat instrument capable of controlling the input parameters of CV and EIS”.

Point 5: In page 8, please use symbols or arrows to differentiate clearly the curves in Figure 7.

Response 5: Symbols were added to the curves in Figures 6, 9, and 12 (previously Figures 7, 10, and 13) to differentiate one curve from another more clearly.

Point 6: In Figure 8, the scale starting in -200 ohms is meaningless.

Response 6: Figure 8 (now Figure 7 due to combination of Figure 1 and 2 into a single figure as suggested by the other reviewer) was rescaled and -200 ohms was removed from the scale.  

Point 7: Please use resistance symbol instead of impedance.

Response 7: EIS Nyquist plots typically use impedance symbol as its x and y axes are components of the impedance Z wherein the x axis corresponds to resistance and y axis describe the energy storage component (in this case, it is the capacitance). Please refer to references in the manuscript for EIS plot notation, particularly reference no.’s 5 & 6.

Point 8: In page 13, the size of the font of the axis label is too strident.

Response 8: Font size of the axis labels of Figure 14 (previously Figure 15) was reduced.

Reviewer 2 Report

I suggest to combine Fig 1 and Fig 2 for a better comparison

The indeces of the crystalline planes on XRD patterns must be shown and discussed in the text.

A more detailed discussion of the physical chemical phoenomena affecting the electrochemical performances could be made.

Actually please explain in more details (from a microscopic/ chemical point of view) why rGO/TNT shows the best behaviour in term of capacity.

The references must be implemented.

Author Response

Response to Reviewer 2 Comments

Point 1: I suggest to combine Fig 1 and Fig 2 for a better comparison.

Response 1: The two figures were combined as suggested.

Point 2: The indeces of the crystalline planes on XRD patterns must be shown and discussed in the text.

Response 2: Miller indices were added in the text and discussed accordingly. Please see discussions on XRD Figures (Fig. 1 and 14) on pp. 3 and 13.

Point 3: A more detailed discussion of the physical chemical phoenomena affecting the electrochemical performances could be made.

Response 3: Please see Results and Discussion section 3.5, p. 12, towards the end of the first paragraph highlighting the effect of presence of TNT during reduction of GO in Method 1 and presence of rGO during TNT formation in Method 2 to the extent of GO reduction and to the morphology of TiO2 respectively.

Point 4: Actually please explain in more details (from a microscopic/ chemical point of view) why rGO/TNT shows the best behaviour in term of capacity.

Response 4: In addition to discussion to satisfy Point 3, a discussion on Figure 15 (SEM and TEM of the best composite) was extended. Please see last paragraph of section 3.5 (p. 14) that points out the microscopic morphology and molecular properties that “… enhanced specific capacitance and minimized redox reactions…”.

Point 5: The references must be implemented.

Response 5: References to support the information and explanations added based on comments were cited accordingly.
